 

# Periaqueductal gray activates antipredatory neural responses in the amygdala of foraging rats

Eun Joo Kim[1], Mi-Seon Kong[2], Sanggeon Park[3,4], Jeiwon Cho[3,4]\*, Jeansok John Kim[1,5]\*

[1]Department of Psychology, University of Washington, Seattle, United States; [2]Department of Psychiatry and Behavioral Sciences, University of Washington, Seattle, United States; [3]Department of Brain and Cognitive Sciences, Scranton College, Ewha Womans University, Seoul, Republic of Korea; [4]Brain Disease Research Institute, Ewha Brain Institute, Ewha Womans University, Seoul, Republic of Korea; [5]Program in Neuroscience, University of Washington, Seattle, United States

**Abstract** Pavlovian fear conditioning research suggests that the interaction between the dorsal periaqueductal gray (dPAG) and basolateral amygdala (BLA) acts as a prediction error mechanism in the formation of associative fear memories. However, their roles in responding to naturalistic predatory threats, characterized by less explicit cues and the absence of reiterative trial-and-error learning events, remain unexplored. In this study, we conducted single-unit recordings in rats during an 'approach food-avoid predator' task, focusing on the responsiveness of dPAG and BLA neurons to a rapidly approaching robot predator. Optogenetic stimulation of the dPAG triggered fleeing behaviors and increased BLA activity in naive rats. Notably, BLA neurons activated by dPAG stimulation displayed immediate responses to the robot, demonstrating heightened synchronous activity compared to BLA neurons that did not respond to dPAG stimulation. Additionally, the use of anterograde and retrograde tracer injections into the dPAG and BLA, respectively, coupled with c-Fos activation in response to predatory threats, indicates that the midline thalamus may play an intermediary role in innate antipredatory-defensive functioning.

\*For correspondence:
jelectro21@ewha.ac.kr (JC);
jeansokk@uw.edu (JJK)

**Competing interest:** The authors declare that no competing interests exist.

## eLife assessment

This study presents **valuable** findings describing how the midbrain periaqueductal gray matter and basolateral amygdala communicate when a predator threat is detected. Though the periaqueductal gray is usually viewed as a downstream effector, this work contributes to a growing body of literature from this lab showing that the periaqueductal gray produces effects by acting on the basolateral amygdala, the experimental design, data collection, and analysis methods provide **solid** evidence for the main claims. The anatomical and immediately early gene evidence that the paraventricular nucleus of the thalamus may serve as a mediator of dorsolateral periaqueductal gray to basolateral amygdala neurotransmission provides an impetus for future functional assessment of this possibility. This study will appeal to a broad audience, including basic scientists interested in neural circuits, basic and clinical researchers interested in fear, and behavioral ecologists interested in foraging.

## Introduction

The interaction between the dorsal periaqueductal gray (dPAG) and the basolateral amygdala (BLA) plays a pivotal role in defensive mechanisms across both animals and humans. Research on Pavlovian fear conditioning (FC), which involves pairing a conditional stimulus (CS; e.g., tones, lights, contexts) with an unconditional stimulus (US; e.g., electric shock) to produce conditional fear responses (CRs), suggests that the dPAG is part of the ascending US pain transmission pathway to the BLA, the presumed site for CS-US association formation (*De Oca et al., 1998*; *Gross and Canteras, 2012*; *Herry and Johansen, 2014*; *Kim et al., 1993*; *Ressler and Maren, 2019*; *Walker and Davis, 1997*). This pathway is thought to be mediated through the lateral and dorsal-midline thalamus regions, including the posterior intralaminar nucleus and paraventricular nucleus of the thalamus (PVT) (*Krout and Loewy, 2000*; *McNally et al., 2011*; *Yeh et al., 2021*; but see *Brunzell and Kim, 2001*). Supporting this model, rodent studies have shown that electrical stimulation of the dPAG, which triggers robust activity bursts (jumping, running, escaping) similar to footshock-induced unconditional responses (URs) (*Jenck et al., 1995*; *Olds and Olds, 1963*), can effectively act as a surrogate US for both auditory and contextual FC in rats (*Di Scala et al., 1987*; *Kim et al., 2013*). In parallel, *Johansen et al., 2010* found that pharmacological inhibition of the PAG, encompassing both dorsal (dPAG) and ventral (vPAG) regions, diminishes the behavioral and neural responses in the amygdala elicited by periorbital shock US, thereby impairing the acquisition of auditory FC. Furthermore, while dPAG neurons initially respond to the shock US, the development of fear CRs results in a reduction of US-evoked neural responses in the PAG due to increased amygdala-PAG pathway-mediated analgesia that dampens footshock nociception (*Johansen et al., 2010*). Fundamentally, the negative feedback circuit between the amygdala and the dPAG serves as a biological implementation of the Rescorla–Wagner model (*Rescorla and Wagner, 1972*), a foundational theory of associative learning that emphasizes the importance of prediction errors in reinforcement (i.e., US), as applied to FC (*Fanselow, 1998*). However, the relevance of PAG–amygdala interactions observed in controlled conditioning chambers remains to be assessed in more realistic threat scenarios.

In human cases of intractable pain surgery, stimulation of the dPAG has been reported to induce intense fear and panic sensations (*Carrive and Morgan, 2012*; *Magierek et al., 2003*). For instance, one patient described the sensation of PAG stimulation as "Something horrible is coming, somebody is now chasing me, I am trying to escape from him" (*Amano et al., 1982*). Similarly, our previous work demonstrated that electrical stimulation of either the dPAG or BLA prompted naïve rats foraging for food in a naturalistic environment to flee to a safe nest in the absence of external threats (*Kim et al., 2013*). Notably, the fleeing behavior induced by dPAG stimulation was contingent on an intact amygdala, whereas BLA stimulation-induced fleeing did not require an intact PAG. This pattern suggests that the fear/panic sensations reported in humans may originate from an innate fear information flow from the dPAG to the BLA.

To further explore this, we employed single-unit recordings and optogenetics in an 'approach food-avoid predator' paradigm (*Choi and Kim, 2010*; *Wang et al., 2015*), focusing on the responsiveness of dPAG and BLA neurons to an approaching robotic predator. Our findings reveal that optogenetic activation of the dPAG not only precipitated escape behaviors but also significantly increased BLA neuron activity. BLA neurons activated through dPAG optogenetic stimulation exhibited immediate and heightened synchronous activity in response to the predator threat, unlike BLA neurons unresponsive to dPAG stimulation. Considering the lack of direct monosynaptic projections between dPAG and BLA neurons (*Vianna and Brandão, 2003*; *McNally et al., 2011*; *Cameron et al., 1995*), we utilized anterograde and retrograde tracers in the dPAG and BLA, respectively. This was complemented by c-Fos expression analysis following exposure to predatory threats. Our anatomical findings suggest that the PVT may be part of a network that conveys predatory threat information from the dPAG to the BLA.

## Results

### dPAG neurons respond to extrinsic predatory threats during a risky foraging task

Five male Long–Evans rats, maintained at 85% of their normal weight through food restriction, were implanted with microdrive arrays positioned above the dPAG. The dPAG includes dorsomedial

(dmPAG), dorsolateral (dlPAG), and lateral (lPAG) regions, as designated by extensive track-tracing, neurochemical, and immunohistochemical studies (e.g., *Bandler et al., 1991*; *Bandler and Keay, 1996*; *Carrive, 1993*). During baseline sessions, rats were required to procure a food pellet in an open arena (202 cm length × 58 cm width × 6 cm height; *Figure 1A*) and return to their nest to consume it. The electrodes were gradually lowered (<120 μm/day) into the dPAG during these sessions (*Figure 1B*). Once well-isolated neural activities were detected, rats underwent predator testing with successive pre-robot, robot, and post-robot sessions (5–15 pellet attempts/session). During the robot session, the animals' outbound foraging time (latency to reach the pellet or the predator-triggering zone) increased, while the pellet success rate decreased (*Figure 1C and D*). We collected a total of 94 dPAG units (*Figure 1—figure supplement 1A and B*) during predator testing, with 23.4% (n = 22) showing increased firing rates ($z > 3$) specifically to the approaching robot with short-latency responses (<500 ms) during the robot session, but not during the pre- and post-robot sessions (robot cells; *Figure 1E and F*, *Figure 1—figure supplement 1C and D*). We focused our analysis on excited robot cells, excluding other types of units that showed food-specific and mixed robot encounter + pellet procurement responses (pellet and BOTH cells; *Figure 1F*, *Figure 1—figure supplement 1A and B*), to investigate predator-related dPAG activity. One cell that exhibited decreased firing rates ($z < -3$) specifically to the robot was also excluded (*Figure 1—figure supplement 1E*). The majority of the robot cells (90.9%) did not show significant correlations between movement speed and neuronal firing rate (*Figure 1—figure supplement 1F*). These results suggest that dPAG neurons participate in detecting an advancing robot and eliciting antipredatory-defensive behaviors, such as fleeing to a safe nest.

## dPAG neurons can intrinsically evoke fear in the absence of external threats

We next investigated if optogenetic activation of dPAG neurons could elicit antipredatory behaviors without an external predator. To do so, we first unilaterally injected rats with ChR2 and implanted an optrode into their dPAG (*Figure 2A*, *Figure 2—figure supplement 1A*). In two anesthetized rats, 20 Hz blue light stimulations (473 nm, 10 ms pulse width, 2 s duration) elicited excited responses in 48% of the recorded units (12 out of 25 cells; *Figure 2B*, *Figure 2—figure supplement 1B*), while only one unit showed inhibited responses to the stimulation (*Figure 2—figure supplement 1C*), confirming the effectiveness of dPAG opto-stimulation. A separate group of rats with unilateral ChR2 (n = 9) or EYFP (n = 4) injections and optic fiber implantation (*Figure 2C*) underwent 4 days of baseline training followed by testing sessions. During testing, rats were allowed to procure a pellet without light stimulation (Off) and with light stimulation (On) (*Figure 2D*). The stimulation parameters employed were identical to those used in the anesthetized rats. During light-on trials (*Figure 2E*), the ChR2-expressing animals were not able to procure the pellet and consistently fled into the nest, whether the pellet was placed at a 76.2 cm long distance from the nest (On$_L$) or at a 25.4 cm short distance from the nest (On$_S$) (*Figure 2F and G*). However, EYFP control rats did not show any defensive behaviors. Latency to procure pellets increased as a function of the stimulation intensity, frequency, and duration (*Figure 2H–J*). These results demonstrate that optical stimulation of CaMKII-expressing dPAG neurons effectively caused naïve rats approaching a food pellet to flee to the nest without an external threat.

## BLA neurons that respond intrinsically to dPAG optical stimulation also respond extrinsically to a robot predator

Our previous data indicated that PAG may transmit innate fear signals to the amygdala (*Kim et al., 2013*). To investigate amygdala neuron responses to dPAG activity changes, we recorded BLA neurons while optically stimulating the dPAG in foraging rats using the same parameters employed in the anesthetized rats during optrode recording (473 nm, 20 Hz, 10 ms pulse width, 2 s duration). Six rats were injected with ChR2 and implanted with an optic fiber in the dPAG and tetrode arrays in the BLA (*Figure 3A*). After the electrodes were lowered to the target structures between the baseline sessions, animals underwent testing comprising pre-stim, stim, and post-stim sessions (*Figure 3B*). The dPAG stimulation increased outbound foraging time (*Figure 3C*) and decreased success rate (*Figure 3D*), mimicking predator-induced cautious behavior (*Figure 1C and D*). While rats were escaping the foraging area without the pellet (stim session), we collected data from 320 BLA neurons. Subsets of

**Figure 1.** Dorsal periaqueductal gray (dPAG) single-unit recordings during risky foraging. (**A**) Rats underwent pre-robot, robot, and post-robot sessions, successfully securing pellets in pre- and post-robot trials, and failing during robot trials due to robot interference. (**B**) Tetrode implantation in dPAG with a photomicrograph of the tip (arrowhead). (**C**) Outbound foraging time increased significantly in the robot session ($X^2 = 64.00$, $p < 0.0001$, Friedman test; $ps < 0.05$ for all comparisons, Dunn's test, n = 42 recording days from 5 rats). ***$p < 0.001$ compared to pre-robot and post-robot sessions. #$p < 0.05$

*Figure 1 continued on next page*

*Figure 1 continued*

compared to the pre-robot session. (**D**) The pellet success rate significantly decreased during robot session ($X^2$ = 84.00, p<0.0001, Friedman test; ps<0.0001 for all comparisons, Dunn's test, n = 42 recording days from 5 rats). ***p<0.001 compared to pre-robot and post-robot sessions. (**E**) Cell-type proportions revealed that 23.4% cells responded to robot activation (robot cells). (**F**) Representative dPAG robot cell raster/event histograms aligned with robot activations. (**G**) Population activity of robot cells around the time of robot activation (t = 0) with 0.1 s and 1 s bins. Firing rates of the robot cells were higher during robot session (0–3 s blocks; Friedman test, all $X^2$s > 6.952, all ps<0.05; Dunn's test, all ps<0.05, n = 22 units). Shaded areas indicate SEM. **p<0.01 compared to pre-robot session. #, ##, and ### denote p<0.05, p<0.01, and p<0.001, respectively, compared to post-robot session. Panel (**E**) created with BioRender.com, and published using a CC BY-NC-ND license with permission.

The online version of this article includes the following figure supplement(s) for figure 1:

**Figure supplement 1.** Dorsal periaqueductal gray (dPAG) unit cell types.

neurons in the BLA (n = 34) exhibited firing increases in response to dPAG stimulation (*Figure 3E*, *Figure 3—figure supplement 1A–C*).

To further investigate how stimulation-responsive neurons respond to an actual predator, three out of the six rats were tested with an approaching robot following the post-stim session (*Figure 3F*). The predatory robot increased outbound foraging time compared to the post-stim session (*Figure 3G*). Among the 85 units recorded in the BLA, robot-specific cells increased firing in response to the advancing robot (robot cells; BLA, 25.9%; *Figure 3—figure supplement 1D–F*). Correlation analysis revealed that 95.4% of BLA robot cells did not show significant correlations between firing rates and movement speed (*Figure 3—figure supplement 1G*). Additionally, 23/85 BLA units were responsive to optical stimulation during the stim session but not pre-/post-stim sessions (*Figure 3H and I*, *Figure 3—figure supplement 1H*). The proportions of the robot and non-robot cells differed between stimulation-responsive and nonresponsive cells, with a lower proportion of robot cells in the stimulation-nonresponsive cells (*Figure 3J*). Stimulation-responsive cells exhibited higher firing rates to the robot predator than stimulation-nonresponsive cells (*Figure 3K*, *Figure 3—figure supplement 1I and J*). The higher the maximal firing rate was during the stimulation, the greater the cells fired to the actual robot predator (*Figure 3L*).

We computed cross-correlations between simultaneously recorded BLA cell pairs to test how subpopulations of BLA neurons co-activate to the predatory threat differently. From 66 BLA recorded neurons, 185 BLA pairs were computed for cross-correlograms (CCs) during post-robot surge (2 s periods subsequent to robot activation; robot session), post-pellet, and post-stim (2 s periods subsequent to pellet procurement; pre- and post-stim sessions) epochs. Twenty-six CCs showed significant peaks (z scores > 3) around the paired spikes (between 0 ms and 100 ms) during the post-surge epoch (*Figure 3—figure supplement 2A*). Cell pairs with spike synchrony during the post-robot surge epoch did not show correlated firing during the post-pellet or post-stim epochs (*Figure 3M*, *Figure 3—figure supplement 2B and C*). This effect was prominent when pairs contained at least one of the stimulation-responsive cells (stim pairs; *Figure 3N*, *Figure 3—figure supplement 2D and E*), but not when pairs include only stimulation-nonresponsive cells (non-stim pairs; *Figure 3N*, *Figure 3—figure supplement 2D and F*). Stim pairs exhibited greater synchronous firing compared to non-stim pairs during the 0–50 ms window but had lower correlated firing during the 50–100 ms window, indicating that dPAG-stimulation responsive cells fire together more closely than dPAG-stimulation nonresponsive cells (*Figure 3O*, *Figure 3—figure supplement 2G*). Additionally, the stim pairs tended to have higher peaks of the CCs than the non-stim pairs (*Figure 3—figure supplement 2H*). Taken together, these results suggest that dPAG stimulation produces activity changes in subpopulations of BLA neurons, primarily in predator detection cells (robot cells), supporting the idea that PAG conveys innate fear signals to the amygdala.

## The midline thalamus interconnects the dPAG and BLA

Based on previous anatomical studies (*Cameron et al., 1995*; *Krout and Loewy, 2000*) and response latency data in the present study (*Figure 3—figure supplement 1B and C*), it is likely that projections from the dPAG to the BLA are indirect. To identify potential mediators that relay predator signals from the dPAG to the BLA, we injected the anterograde tracer AAV-CaMKII-EYFP into the dPAG and investigated robot-induced c-Fos activities in AAV-expressed terminal areas of the dPAG. Cholera toxin



**Figure 2.** Dorsal periaqueductal gray (dPAG) optical stimulation evokes fear. (**A**) Virus injection, optrode implantation in dPAG, and light stimulation during single-unit recordings in anesthetized rats. (**B**) Raster plots and peri-event time histograms for 20 Hz light stimulations (10 ms width, left; 2 s duration, center). 48% of 25 units had increased firing during 2 s light stimulation (right). (**C**) Virus injection, expression, and optic fiber placement in dPAG. (**D**) Stimulation testing: baseline trials at 75 cm distance (long) without light; stimulation trials with 2 s light as the rat approached (~25 cm) long pellet; light applied as the rat approached 25 cm (short) pellet if long pellet unsuccessful. (**E**) Representative trajectories for EYFP- and ChR2-expressing rats during stimulation testing. (**F**) Rat behaviors during light stimulation trials. (**G**) ChR2 (n = 9) rats showed increased latency to procure pellet upon opto-stimulation (On$_L$) compared to EYFP (n = 4) rats (Off$_L$, $Z$ = 1.013, p=0.311; On$_L$, $U$ = 0.0, p<0.001; Mann–Whitney $U$ test). (**H–J**) ChR2 group (n = 9) exhibited increased latency to procure pellet compared to EYFP group (n = 4) based on stimulation intensity (**H**; $U$s <for all intensities < 3.5, ps for all intensities <0.025; Mann–Whitney $U$ test), frequency (**I**; $U$s for 10 Hz and 20 Hz <2.5, ps for 10 and 20 Hz <0.014; Mann–Whitney U test), and duration (**J**; $U$s for all durations <4.5, ps for all durations <0.032; Mann–Whitney $U$ test). *, **, and *** denote p<0.05, p<0.01, and p<0.001, respectively. Panel **D** created with BioRender.com, and published using a CC BY-NC-ND license with permission.

*Figure 2 continued on next page*

2000

The online version of this article includes the following figure supplement(s) for figure 2:

**Figure supplement 1.** Optrode recording in the dorsal periaqueductal gray (dPAG).

subunit B (CTB) was also injected into the BLA in a subset of AAV-injected animals to examine potential anatomical evidence of relays between the dPAG and BLA (*Figure 4A and B*). After recovery, rats were trained to procure a pellet from the foraging arena. Animals were randomly assigned to foraging-only and robot-experienced groups. The foraging-only group quickly procured the pellet, while the robot-experienced group failed during the testing session (*Figure 4C*). Ninety minutes after testing, animals were perfused to analyze c-Fos activity and the tracers. Consistent with prior research (*Cameron et al., 1995*; *Krout and Loewy, 2000*), we also observed strong terminal expression in midline thalamic areas emerging from dPAG cell body AAV infection (*Figure 4D*). Among these, the PVT showed increased levels of c-Fos-positive cells in rats encountering the robot predator compared to controls (*Figure 4E and F*). In animals that were also injected with CTB in the BLA, the robot encounter increased the percentage of CTB-labeled PVT neurons expressing c-Fos, compared to the foraging-only control, while the density of CTB remained comparable between the groups (*Figure 4G and H*, *Figure 4—figure supplement 1*). Collectively, these results suggest predator-induced dPAG activity may convey information to the BLA, potentially through PVT activity.

## Discussion

Prehistoric rock carvings and cave drawings of large carnivores from the Ice Age suggest that the fear of predation played a crucial role in the lives of early humans in their habitats (*Mithen, 1999*). As such, the brain's fear system, which evolved to counteract unpredictable, uncontrolled threats originating from biological agents, likely influences our everyday behavior when confronted with perceived risks. In line with this view, studies using functional magnetic resonance imaging (fMRI) and an 'active escape paradigm'—in which virtually represented subjects are chased by a virtual predator and receive an electric shock when 'caught'—have shown that as the threat transitions from a distant position to an impending encounter with the subject, neural activity shifts from the ventromedial PFC and amygdala toward the PAG (*Mobbs et al., 2009*; *Mobbs et al., 2007*).

The present study investigated the neural mechanisms underlying antipredatory-defensive behaviors in rats engaged in a naturalistic goal-directed behavior of foraging for food in an open arena, using a combination of electrophysiology, optogenetics, and tracing techniques. We found that neurons in the dPAG are involved in detecting predatory threats and eliciting antipredatory-defensive behaviors in rats. Specifically, dPAG neurons displayed increased spiking in response to an approaching robot predator as rats reflexively fled from the open foraging arena into a safe nest. However, dPAG neuronal activity did not increase when the robot was stationary, which contrasts with a previous study that reported enhanced dPAG neuronal activity in mice exposed to an anesthetized (motionless) rat separated by a wire mesh in a chamber (*Deng et al., 2016*) or a live rat tethered (restrained) with a harness (*Reis et al., 2021*). It has been suggested that looming stimuli may serve as simple, evolutionarily reliable signals of danger because genes are not capable of providing the brain with detailed information about all potential predatory threats, and all predators must approach their prey for consumption (*Kim and Jung, 2018*). Our study also demonstrated that optogenetic stimulation of CaMKII-expressing dPAG neurons caused naïve rats approaching a food pellet to instantly flee to the nest without an external threat (see also *Tsang et al., 2023*). This confirmed escape behavior in foraging rats with dPAG electrical stimulation (*Kim et al., 2013*) was evoked by intrinsic dPAG neurons, not the fibers of passage or current spread to other brain regions. In contrast to our observation of escape-to-the-nest behavior, optogenetic activation of dPAG in mice elicited various defensive behaviors, such as running, freezing, and conditioned avoidance, when tested in a chamber (*Deng et al., 2016*). This is consistent with the notion that the behavioral outcome from brain stimulation is influenced by environmental settings (*Kim et al., 2013*).

As mentioned in the introduction, the dPAG is recognized as part of the ascending nociceptive pathway to the BLA (*De Oca et al., 1998*; *Gross and Canteras, 2012*; *Herry and Johansen, 2014*;



**Figure 3.** Dorsal periaqueductal gray (dPAG) optical stimulation and amygdala recordings. (**A**) Virus injection in dPAG and tetrode array implantation targeting basolateral amygdala (BLA). Light stimulation during single-unit recordings in freely moving rats. (**B**) Stimulation testing sessions: in pre- and post-stim trials, rats freely procured pellets; in stim trials, optical stimulation prevented procurement of pellets. (**C, D**) During dPAG stimulation, animals showed increased outbound foraging time (**C**; $X^2$ = 117.8, p<0.0001, Friedman test; ps<0.0001, Dunn's test, n = 78 recording days from 7 rats) and decreased success rate (**D**; $X^2$ = 154.0, p<0.0001, Friedman test; ps<0.0001, Dunn's test, n = 78 recording days from 7 rats). ****p<0.0001 compared to pre-robot and post-robot sessions. ####p<0.0001 compared pre-robot session. (**E**) Subset of BLA units (10.0%) responsive to optical stimulation (Stim cells; left), and a representative (center) and all stimulation-responsive (Stim cells; right; n = 320 cells) raster plots with peri-event time histograms (PETHs). (**F**) Subset of animals (n = 3 rats) underwent additional robot trials following the post-stim session. (**G**) Increased outbound foraging time during robot session compared to post-stimulation session (t(15) = 6.655, p<0.0001; paired t-test; 16 recording days from 3 rats). ****p<0.0001. (**H**) Twenty-two BLA units were dPAG stimulation-responsive. (**I**) Representative raster plots of dPAG stimulation-responsive and -nonresponsive units. (**J**) Proportions of robot vs. non-robot cells differed between stimulation-responsive and -nonresponsive units ($X^2$ = 11.134, p<0.001; chi-square test). ***p<0.001. (**K**) PETHs of stim (n = 23 cells) and non-stim cells (n = 62 cells) during stimulation and robot sessions. (**L**) Relationship between maximal firing rates during first 500 ms subsequent to robot activation and maximal firing rates during first 500 ms after stimulation onset (r(85) = 0.405, p<0.001; Pearson correlation). (**M**) Population cross-correlograms (CCs) with significant synchrony during robot session (n = 26 CCs) were higher than other sessions. Dotted vertical lines indicate 0–100 ms window for testing significance. Gray, blue, and dark yellow ***p<0.001 compared to pre-stimulation, stimulation,

*Figure 3 continued on next page*

*Figure 3 continued*

and post-stimulation sessions, respectively. (**N**) Among synchronized BLA cell pairs, those including dPAG stimulation-responsive cell(s) (stim pairs; 61.5%; n = 16 pairss) showed increased correlated firings (area under the curve [AUC] during 0–100 ms window) during the robot session compared to other sessions. In contrast, synchronized BLA cell pairs that consisted of stimulation nonresponsive cells only (n = 10 pairs) showed no AUC differences across sessions. Gray *, blue **, and dark yellow **p<0.05 compared to pre-stimulation, p<0.01 compared to stimulation, and p<0.001 compared to post-stimulation sessions, respectively. (**O**) Comparing CCs during testing windows (0–50 ms and 50–100 ms) between stim (n = 16) and non-stim pairs (n = 10), stim pairs exhibited higher correlated firing than non-stim pairs during the 0–50 ms block (t(21.99) = 2.342, p=0.0286; t-test), while displaying decreased correlated firings in the second block (50–100 ms; U = 42, p=0.045; Mann–Whitney U test). *p<0.05 compared to the non-stim pairs. Panels **B** and **F** created with BioRender.com, and published using a CC BY-NC-ND license with permission.

The online version of this article includes the following figure supplement(s) for figure 3:

**Figure supplement 1.** Characteristics of basolateral amygdala (BLA) cells.

**Figure supplement 2.** Spike synchrony in dorsal periaqueductal gray (dPAG)-stimulated basolateral amygdala (BLA) neurons under predatory threats.

*Kim et al., 1993*; *Ressler and Maren, 2019*; *Walker and Davis, 1997*). The dPAG is also implicated in non-opioid analgesia (*Bagley and Ingram, 2020*; *Cannon et al., 1982*; *Fields, 2000*). However, it is essential to emphasize that, despite its roles in pain modulation, the primary behavior observed in dPAG-stimulated, naive rats foraging for food in an open arena was goal-directed escape to the safe nest, underscoring the dPAG's critical function in survival behaviors.

Our study also revealed that BLA neurons respond both to direct dPAG optical stimulation and to an external robot predator. Specifically, BLA cells responsive to dPAG stimulation showed increased firing rates and were more likely to be co-activated by predatory threats compared to dPAG stimulation-non-responsive BLA cells. Additionally, the firing rates of BLA cells in response to dPAG stimulation highly correlated with their responses to the actual predatory agent, indicating that BLA neurons involved in processing dPAG signals may play a crucial role in enhancing the coherence of the BLA network, thereby effectively managing predatory threat information. These findings, combined with previous research demonstrating that the amygdala is necessary for both intrinsic dPAG stimulation-induced (*Kim et al., 2013*) and extrinsic robot-evoked (*Choi and Kim, 2010*) escape behavior, suggest the dPAG-amygdala pathway's involvement in processing innate fear signals and generating antipredatory-defensive behavior. However, it is crucial to consider the recent discovery that optogenetic stimulation of CA3 neurons (3000 pulses) leads to gain-of-function changes in CA3-CA3 recurrent (monosynaptic) excitatory synapses (*Oishi et al., 2019*). Although there is no direct connection between dPAG neurons and the BLA (*Vianna and Brandão, 2003*; *McNally et al., 2011*; *Cameron et al., 1995*), and no studies have yet demonstrated gain-of-function changes in polysynaptic pathways to our knowledge, the potential for our dPAG photostimulation (320 pulses) to induce similar changes in amygdalar neurons, thereby enhancing their sensitivity to predatory threats, cannot be dismissed.

We confirmed strong terminal expressions in the midline thalamic areas (*Cameron et al., 1995*; *Krout and Loewy, 2000*) following AAV injection into the dPAG using immunohistochemistry. Notably, the PVT, in contrast to other midline thalamic subregions, exhibited increased c-Fos reactivity in response to the robot predator. This observation aligns with recent findings that dPAG projections specifically target the PVT rather than the centromedial intralaminar thalamic nucleus (*Yeh et al., 2021*). Additionally, subpopulations of the c-Fos-reactive PVT neurons were found to express CTB, retrogradely labeled by injection into the BLA. These results, in conjunction with previous research on the roles of the dPAG, PVT, and BLA in producing flight behaviors in naïve rats (*Choi and Kim, 2010*; *Daviu et al., 2020*; *Deng et al., 2016*; *Kim et al., 2013*; *Kim et al., 2018*; *Kong et al., 2021*; *Ma et al., 2021*; *Reis et al., 2021*), the anterior PVT's involvement in cat odor-induced avoidance behavior (*Engelke et al., 2021*), and the PVT's regulation of behaviors motivated by both appetitive and aversive stimuli (*Choi et al., 2019*; *Choi and McNally, 2017*), suggest the involvement of the dPAG→PVT→BLA pathways in antipredatory-defensive mechanisms, particularly as rats leave the safety of the nest to forage in an open arena (*Figure 4I*; *Reis et al., 2023*). Given the recent significance attributed to the superior colliculus in detecting innate visual threats (*Lischinsky and Lin, 2019*;



**Figure 4.** Paraventricular nucleus of the thalamus' (PVT) hypothesized role in dorsal periaqueductal gray (dPAG) to amygdala signaling and antipredatory behavior model. (**A**) Cholera toxin subunit B (CTB) retrograde tracer and AAV-CaMKII-EYFP used to trace dPAG signals to the basolateral amygdala (BLA). (**B**) Representative images of AAV and CTB expressions in dPAG and BLA, respectively. (**C**) Robot encounters hindered pellet procurement compared to foraging-only rats (Base, $U = 26$, p=0.6926; Test, $U = 0$, p=0.0001; Mann–Whitney $U$ test). ***p<0.001 compared to

*Figure 4 continued on next page*

*Figure 4 continued*

the foraging-only group. (**D**) Terminal expressions of AAV injected into the dPAG cell bodies were predominantly observed in the midline nuclei of the thalamus. (**E**) PVT showed higher c-Fos-positive cells in robot-experienced rats (n = 10) compared to foraging-only control rats (n = 6) (*U* = 4.0, p=0.0027; Mann–Whitney *U* test), while other midline thalamic areas showed no differences (*t*(14)s < 1.611, ps>0.129; *t*-test). **p<0.01 compared to the foraging-only group. Values were normalized to the mean of the corresponding control group. (**F**) Representative photomicrographs of PVT c-Fos staining from foraging-only (upper) and robot-experienced (bottom) rats. (**G**) Representative microphotographs of AAV, CTB, c-Fos, and triple staining in PVT comparing foraging-only and robot-experienced animals. (**H**) Robot exposure increased the percentage of CTB-labeled PVT neurons expressing c-Fos (*t*(9) = 3.171, p=0.0113), while CTB density levels were comparable between the two groups (robot-experienced, n = 5; foraging-only, n = 6; *t*(9) = 0.9039, p=0.3896; *t*-test). Values were normalized to the mean of the corresponding control group. (**I**) Proposed model: predator surge detection via visual pathways, e.g., superior colliculus (*Furigo et al., 2010*; *Rhoades et al., 1989*), leads to dPAG activation, signaling through PVT to excite BLA. The BLA then projects to regions controlling escape responses, such as dorsal/posterior striatum (*Li et al., 2021*; *Menegas et al., 2018*) and ventromedial hypothalamus (*Silva et al., 2013*; *Kunwar et al., 2015*; *Wang et al., 2015*).

The online version of this article includes the following figure supplement(s) for figure 4:

**Figure supplement 1.** Double labeling of cholera toxin subunit B and c-FOS in paraventricular nucleus of the thalamus (PVT) cells.

---

*Wei et al., 2015*; *Zhou et al., 2019*) and the cuneiform nucleus in the directed flight behavior of mice (*Bindi et al., 2023*; *Tsang et al., 2023*), further exploration into the communication between these structures and the dPAG-BLA circuitry is warranted.

While our findings demonstrate that opto-stimulation of the dPAG is sufficient to trigger both fleeing behavior and increased BLA activity, we have not established that the dPAG-PVT circuit is necessary for the BLA's response to predatory threats. To establish causality and interregional relationships, future studies should employ methods such as pathway-specific optogenetic inhibition (using retrograde Cre-recombinase virus with inhibitory opsins; *Lavoie and Liu, 2020*; *Li et al., 2016*; *Senn et al., 2014*) or chemogenetics (*Boender et al., 2014*; *Roth, 2016*) in conjunction with single-unit recordings to fully characterize the dPAG-PVT-BLA circuitry's (as opposed to other midline thalamic regions for controls) role in processing predatory threat-induced escape behavior. If inactivating the dPAG-PVT circuits reduces the BLA's response to threats, this would highlight the central role of the dPAG-PVT pathway in this defense mechanism. Conversely, if the BLA's response remains unchanged despite dPAG-PVT inactivation, it could suggest the existence of multiple pathways for antipredatory defenses.

Future studies will also need to delineate the downstream pathways emanating from the BLA that orchestrate goal-directed flight responses to external predatory threats as well as internal stimulations from the dPAG/BLA circuit. Potential key structures include the dorsal/posterior striatum, which has been associated with avoidance behaviors in response to airpuff in head-fixed mice (*Menegas et al., 2018*) and flight reactions triggered by auditory looming cues (*Li et al., 2021*). Additionally, the ventromedial hypothalamus (VMH) has been implicated in flight behaviors in mice, evidenced by responses to the presence of a rat predator (*Silva et al., 2013*) and upon optogenetic activation of VMH Steroidogenic factor 1 (*Kunwar et al., 2015*) or the VMH-anterior hypothalamic nucleus pathway (*Wang et al., 2015*). Investigating the indispensable role of these structures in flight behavior could involve lesion or inactivation studies. Such interventions are anticipated to inhibit flight behaviors elicited by amygdala stimulation and predatory threats, confirming their critical involvement. Conversely, activating these structures in subjects with an inactivated or lesioned amygdala, which would typically inhibit fear responses to external threats (*Choi and Kim, 2010*), is expected to induce fleeing behavior, further elucidating their functional significance.

The innate predator-fear functions of the dPAG and BLA are distinct from their learned fear functions, as observed in FC studies. In FC, the dPAG is proposed to play a role in 'US-processing', transmitting footshock (pain) information to the amygdala, thus enhancing the CS pathway's ability to activate the amygdalar fear system (*Fanselow, 1998*; *Maren, 2001*; *Almeida et al., 2004*). This hypothesis is supported by research demonstrating that muscimol inactivation of the dPAG hinders FC to the footshock US (*Johansen et al., 2010*), while electrical stimulation of dPAG neurons effectively serves as a US surrogate for FC (*Ballesteros et al., 2014*; *Kim et al., 2013*). As conditioning progresses, amygdalar fear may suppress footshock-evoked dPAG's teaching signals, thereby moderating FC in the amygdala, potentially through conditioned analgesia that dampens the footshock US's painfulness (*Fanselow, 1981*; *Fanselow and Bolles, 1979*). Consistent with this amygdalar-dPAG negative feedback notion, research has revealed that as CS-evoked responses of amygdalar neurons

increase with CS-US pairings, footshock-evoked neural activity in the dPAG decreases (*Johansen et al., 2010*; *Ozawa et al., 2017*; *Quirk et al., 1995*). It remains to be determined whether there are distinct populations of innate predator-responsive and pain-responsive dPAG neurons or if the same dPAG neurons react to a wide range of aversive stimuli. Furthermore, it is worth noting that the same dPAG stimulation resulted in an activity burst UR in a small operant chamber, as opposed to a goal-directed escape to a nest UR in a large foraging arena (*Kim et al., 2013*). Additionally, FC to shock or predator US does not readily occur in naturalistic settings (*Zambetti et al., 2022*).

Overall, this study enhances our understanding of the neural basis of antipredatory-defensive behaviors in rats, emphasizing the important roles of the dPAG and BLA in processing and responding to threat-related stimuli. The study also suggests the possibility of an innate fear mechanism, which may complement the commonly studied learned fear mechanism in the research of anxiety and fear-related disorders. Further research into these mechanisms could lead to the development of novel therapeutic interventions that target both innate and learned fear mechanisms, improving the treatment of anxiety and fear-related disorders.

## Materials and methods
### Subjects
Male and female Long–Evans rats (initial weight 325–350 g) were individually housed in a climate-controlled vivarium (accredited by the Association for Assessment and Accreditation of Laboratory Animal Care), with a reversed 12 hr light/dark cycle (lights on at 7 PM) and placed on a standard food-restriction schedule with free access to water to gradually reach ~85% normal body weights. All experiments were performed during the dark cycle in compliance with the University of Washington Institutional Animal Care and Use Committee guidelines (IACUC protocol #: 4040-01).

### Surgery
Under anesthesia (94 mg/kg ketamine and 6 mg/kg xylazine, intraperitoneally), rats were mounted in a stereotaxic instrument (Kopf) and implanted with a Microdrive of tetrode bundles (formvar-insulated nichrome wires, 14 μm diameter; Kanthal) into the dPAG (AP, –6.8; ML, +0.6; from Bregma), or BLA (AP, –2.8; ML, +5.2; from Bregma). For optogenetic stimulation, anesthetized male rats were injected with adeno-associated viruses (AAVs; serotype 5) to express Channelrhodopsin-EYFP (AAV5-CaMKIIa-hChR2(H134R)-EYFP, UNC Vector Core) or EYFP only (AAV5-CaMKIIa-EYFP) in the dPAG at a rate of 0.05 μl/min (total volume of 0.5 μl) via a microinjection pump (UMP3-1, World Precision Instruments) with a 33-gauge syringe (Hamilton). Animals were randomly assigned to each group. To avoid back-flow of the virus, the injection needle was left in place for 10 min. After virus injection, an optic fiber attached to ferrules (0.22 NA, 200 μm core; ferrule diameter: 2.5 mm; Doric Lenses) or an optrode was implanted 0.4 mm dorsal to the injection sites. The microdrive and/or ferrule were secured by Metabond and dental cement with anchoring screws. Behavioral and recording experiments started after at least 1 week of recovery while optogenetic stimulation sessions started after at least 4 weeks after the surgery to allow for sufficient viral expression. For tracing experiments, male and female rats were injected with AAV-CamKII-EYFP into the dPAG and CTB into the BLA > 5 weeks and 1 week prior to the predator testing, respectively.

### Foraging apparatus
A custom-built foraging arena comprised a nest (29 cm × 57 cm × 60 cm height) and a foraging area (202 cm length × 58 cm width × 6 cm height) with a V-shaped gate connecting the two areas. The animals' movement was automatically tracked via two video tracking systems (Neuralynx and Any-maze).

### Behavioral procedures
#### Habituation
Animals were placed in the nest area, where they were acclimated to the experimental room and chamber and allowed to eat food pellets (0.5 g; F0171, Bio-Serv) for 30 min/day for two consecutive days.

## Baseline foraging

After the animals were placed in the nest, the gate to the foraging area was opened, allowing them to explore the arena. The animals were gradually trained to acquire a food pellet from various distances (25 cm, 50 cm, and 75 cm from the nest for the behavior-only experiment; an additional 100 cm and 125 cm for the recording experiment) and then return to their nest to consume it. Once the rat returned to the nest with the pellet, the gate was closed. The behavior-only groups underwent 4 days of baseline foraging training. For the recording experiments, baseline sessions continued until unit activity in the dPAG or amygdala unit was successfully detected.

## Testing

For dPAG recordings, single units were recorded across three successive sessions: pre-robot, robot, and post-robot (7–14 trials/session; 4–14 recording days; n = 5 rats, 96 units). The number of trials varied among subjects because trials were excluded if rats attempted to procure the pellet within 10 s post-dPAG stimulation or robot activation. This exclusion criterion was implemented to ensure an accurate characterization of unit responsiveness. In the pre- and post-robot trials, rats were permitted to freely procure the pellet. Before initiating the robot trials, a programmed robot (LEGO Mindstorms EV3 set) was placed at the end of the foraging arena. Upon the gate's opening, whenever the rat came near (~25 cm) the pellet, the programmed robot would surge forward 23 cm, 60 cm, or 140 cm toward the animal at a velocity of ~75 cm/s, before returning to its original position. If the rat made another attempt to procure the pellet within 10 s after a robot activation, both the preceding and subsequent trials were excluded from the analyses to isolate and measure the unit properties accurately. For optical stimulation and behavioral experiments, the procedure included three baseline trials with the pellet placed 75 cm away, followed by three dPAG stimulation trials with the pellet locations sequentially set at 75 cm, 50 cm, and 25 cm. During each approach to the pellet, rats received 473 nm light stimulation (1–2 s, 20 Hz, 10 ms width, 1–3 mW) through a laser (Opto Engine LLC) and a pulse generator (Master-8; A.M.P.I.). Additional testing to examine the functional response curves was conducted over multiple days, with incremental adjustments to the stimulation parameters (intensity, frequency, duration) after confirming that normal baseline foraging behavior was maintained. For these tests, one parameter was adjusted incrementally while the others were held constant (intensity curve at 20 Hz, 2 s; frequency curve at 3 mW, 2 s; duration curve at 20 Hz, 3 mW). If the rat failed to procure the pellet within 3 min, the gate was closed, and the trial was concluded. For amygdala recordings with optical stimulation of the dPAG, pre-stim, stim, and post-stim sessions (5–15 trials/session) were carried out successively on the same day. These three consecutive phases mirrored the pre-robot, robot, and post-robot sessions, except that the dPAG light stimulation was given instead of the robot predator (6–18 recording days; n = 6 rats, 320 units). A subset of these animals (n = 3 out of 6 rats, 85 out of 320 units) also participated in a robot session following the post-stim session on selected days (1–16 recording days). Sessions involving the robot predator were repeated until habituation was observed or recordings were deemed invalid due to limitations with the microdrive or failure to detect units. Note that the mean success rate for the dPAG recordings was 2.8% ± 1.31, and animals subjected to the BLA recordings did not succeed in retrieving pellets during any of the robot trials.

## Optrode recording under anesthesia

Upon optrode implantation, the animal was kept under anesthesia and moved to a unit recording setup (Neuralynx). The electrode was slowly lowered, and dPAG activity was monitored. Once dPAG activity was detected, eight light stimulations (2 s, 20 Hz, 10 ms width) were given with 30 s inter-stimulus intervals. In total, 10 and 14 stimulation sessions were repeated in the two anesthetized rats, respectively. After the recording was completed, the animal was immediately perfused under an overdose of Beuthanasia. Only the unit data collected in the dPAG were included in the analyses after histological verification.

## Single-unit recording and analyses

Extracellular single-unit activity was recorded through a 24-channel microdrive array loaded with a bundle of six tetrodes. The electrode tip was gold-plated to 100–300 kΩ measured at 1 kHz. The signals were amplified (×10,000), filtered (600–6000 kHz), and digitized (32 kHz) using a Cheetah data acquisition system (Neuralynx). A spike-sorting program (SpikeSort 3D; Neuralynx) and additional

manual cutting were used for cluster isolation. Peri-event time histograms (PETH) were generated using NeuroExplorer (version 5.030; Nex Technologies) and further analyzed with custom MATLAB programs. The unit and speed data were binned at 0.1 s or 1 s and aligned to the time when the robot was activated, when the pellet was procured, or when the rat turned its body immediately before fleeing to the nest. All PETH data were normalized (z-scored) to the pre-event baseline period (from −5 to 0 s prior to pellet procurement or robot activation, across 50 bins). Food approaching was indicated by the animal entering a designated zone 19 cm from the pellet, a behavior driven by hunger. To classify the responsiveness of the unit responses, the first five bins (0.1 s bins) were analyzed. If one or more bins within the 500 ms period showed z-scores higher than 3 (z > 3) exclusively during the robot session, then the unit was classified as a 'robot cell'. The rest of the cells were further classified as food-specific (z > 3 exclusively during pre-robot session), BOTH (z > 3 during both pre-robot and robot sessions), or none (non-responsive) cells. In the light stimulation and recording experiments, the same criteria were applied to the unit classification except that the stimulation-responsive units (stim cells) were defined as cells showing significant activity (z > 3) during the stim session when aligned to the stimulation onset instead of the robot activation.

## Cross-correlation

BLA cells that were simultaneously recorded from rats undergoing the four successive sessions (pre-stim, stim, post-stim, and robot sessions) were analyzed to generate CCs. Units firing less than 0.1 Hz were excluded due to the possible false peaks in their CCs. The shift predictors (100 random trial shuffles) were subtracted from the raw CCs. The correlated firing (in 10 ms bins) during the 0–2 s period following robot activation was calculated, and the CCs that showed significant peaks (z > 3) within the 0–100 ms window during the robot session were further analyzed. The peak values of the CCs and the peak areas under CC curves during the 100 ms window were compared across sessions (pre-stim, stim, post-stim, and robot sessions).

## Histology

To verify the electrode placement, rats were overdosed with Beuthanasia, given electrolytic currents (10 µA, 10 s) in the target regions through the tetrode tips, and perfused intracardially with 0.9% saline and then 10% formalin. For rats injected with the virus, phosphate-buffered saline (PBS) and 4% paraformaldehyde were used as perfusates. Extracted brains were stored in the fixative at 4°C overnight followed by 30% sucrose solution until they sank. Transverse sections (50 µm) were mounted on gelatin-coated slides and stained with cresyl violet and Prussian blue dyes to examine the recording sites. To confirm viral expression, 30 µm sections were cut, mounted, and cover-slipped with Flouro-mount-GTM with DAPI (eBioscience). Immunohistochemistry was performed on some of the sections to visualize the viral and c-Fos expressions. In short, the sections were washed with 0.1 M PBS for 10 min three times, followed by three rinses with PBS with Triton X-100 (PBST). After 2 hr of blocking (using normal goat serum), the sections were incubated in primary antibodies (1:500 mouse anti-GFP and rabbit anti-c-Fos; Abcam) overnight. The following day, additional PBST washes and secondary antibodies (1:250 anti-mouse Alexa 488 and anti-rabbit Alexa 405; Abcam) were applied. The sections were examined under a fluorescence microscope (Keyence BZ-X800E) and analyzed using ImageJ (NIH). Only rats with correct viral expression or electrode/optic fiber tip locations in the target structure(s) were included for the statistical analyses (*Supplementary file 1A*).

## Statistical analyses

Based on normality tests (Kolmogorov–Smirnov test, p<0.01; *Supplementary file 1B*), non-normally distributed variables were analyzed using non-parametric tests, while parametric tests were applied to normally distributed variables. Statistical significance across sessions was determined using the Friedman test, followed by Dunn's multiple-comparison test with correction when needed. Pearson's correlation coefficients evaluated variable relationships, and chi-square tests compared percentages of distinct unit types. Group comparisons involved independent t-tests or Mann–Whitney U tests, and within-group comparisons employed paired t-tests or Wilcoxon signed rank tests. Statistical analyses and graph generation were performed using SPSS (ver. 19), custom MATLAB codes, GraphPad Prism (ver. 9.00), and NeuroExplorer (ver. 5.030).

## Acknowledgements

This study was supported by the National Institutes of Health grants MH099073 (JJK), AG067008 (EJK), and F32MH127801 (MK), and the Ministry of Science and ICT through the National Research Foundation of Korea grant: Brain Science Research Program NRF-2022M3E5E8018421 (JC) and NRF-2022R1A2C2009265 (JC). Artworks in Figures 1E, 2D, 3B and F were created with BioRender.com, and published using a CC BY-NC-ND license with permission.

## Additional information

### Funding

| Funder | Grant reference number | Author |
| --- | --- | --- |
| National Institutes of Health | MH099073 | Jeansok John Kim |
| National Institutes of Health | F32MH127801 | Mi-Seon Kong |
| National Research Foundation of Korea | NRF-2022M3E5E8018421 | Jeiwon Cho |
| National Research Foundation of Korea | NRF-2022R1A2C2009265 | Jeiwon Cho |
| National Institutes of Health | AG067008 | Eun Joo Kim |

The funders had no role in study design, data collection and interpretation, or the decision to submit the work for publication.

### Author contributions

Eun Joo Kim, Conceptualization, Data curation, Formal analysis, Funding acquisition, Validation, Investigation, Visualization, Methodology, Writing – original draft, Project administration, Writing – review and editing; Mi-Seon Kong, Data curation, Funding acquisition, Validation, Investigation, Methodology, Writing – review and editing; Sanggeon Park, Data curation, Software, Formal analysis, Validation, Visualization, Methodology, Writing – review and editing; Jeiwon Cho, Jeansok John Kim, Conceptualization, Supervision, Funding acquisition, Validation, Methodology, Writing – original draft, Project administration, Writing – review and editing

### Author ORCIDs

Eun Joo Kim ⓘ https://orcid.org/0000-0002-8499-9135
Mi-Seon Kong ⓘ https://orcid.org/0000-0001-8970-7034
Sanggeon Park ⓘ https://orcid.org/0000-0003-2083-2536
Jeiwon Cho ⓘ https://orcid.org/0000-0001-6903-3562
Jeansok John Kim ⓘ https://orcid.org/0000-0001-7964-106X

### Ethics

All experiments were performed during the dark cycle in compliance with the University of Washington Institutional Animal Care and Use Committee guidelines (IACUC protocol: #4040-01).

Reviewer #1 (Public Review): https://doi.org/10.7554/eLife.88733.4.sa1
Reviewer #2 (Public Review): https://doi.org/10.7554/eLife.88733.4.sa2
Reviewer #3 (Public Review): https://doi.org/10.7554/eLife.88733.4.sa3
Author response https://doi.org/10.7554/eLife.88733.4.sa4

## Additional files

### Supplementary files

• Supplementary file 1. Reconstructions of histology and normality test results. (**A**) Histological

reconstructions of recording sites in the dPAG and BLA, and optic fiber locations in the dPAG. (A) Red bars show the trajectory of tetrode recording sites in the dPAG. (B) Red bars depict the trajectory of optrode recording sites in the dPAG. (B) Red and gray circles represent the optic fiber locations for ChR2 and EYFP rats, respectively. (D) Orange circles and red bars indicate the optic fiber locations in the dPAG (left) and recording trajectories in the BLA (right), respectively. Numerical values represent AP coordinates relative to Bregma. (**B**) Normality test. The normality of the variable distributions was assessed using the Kolmogorov–Smirnov test (p<0.01). Depending on the results of this test, parametric tests were used for normally distributed variables, while nonparametric tests were employed for variables that were not normally distributed.

- MDAR checklist

## Data availability

The data that support the findings of this study is available from the Dryad data repository at https://doi.org/10.5061/dryad.stqjq2cbh. The customized analysis tools are deposited on GitHub: https://github.com/KimLab-UW/Velocity (copy archived at *Park, 2024*).

The following dataset was generated:

| Author(s) | Year | Dataset title | Dataset URL | Database and Identifier |
|---|---|---|---|---|
| Kim EJ, Kong MS, Park S, Cho J, Kim JJ | 2024 | Periaqueductal gray activates antipredatory neural responses in the amygdala of foraging rats | https://doi.org/10.5061/dryad.stqjq2cbh | Dryad Digital Repository, 10.5061/dryad.stqjq2cbh |

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
