## [Editor Report · eLife assessment]

This study presents **valuable** findings describing how the midbrain periaqueductal gray matter and basolateral amygdala communicate when a predator threat is detected. Though the periaqueductal gray is usually viewed as a downstream effector, this work contributes to a growing body of literature from this lab showing that the periaqueductal gray produces effects by acting on the basolateral amygdala, the experimental design, data collection, and analysis methods provide **solid** evidence for the main claims. The anatomical and immediately early gene evidence that the paraventricular nucleus of the thalamus may serve as a mediator of dorsolateral periaqueductal gray to basolateral amygdala neurotransmission provides an impetus for future functional assessment of this possibility. This study will appeal to a broad audience, including basic scientists interested in neural circuits, basic and clinical researchers interested in fear, and behavioral ecologists interested in foraging.

---

## [Referee Report · Reviewer #1 (Public Review)]

In the presence of predators, animals display attenuated foraging responses and increased defensive behaviors that serve to protect them from potential predatory attacks. Previous studies have shown that the basolateral nucleus of the amygdala (BLA) and the periaqueductal gray matter (PAG) are necessary for the acquisition and expression of conditioned fear responses. However, it remains unclear how BLA and PAG neurons respond to predatory threats when animals are foraging for food. To address this question, Kim and colleagues conducted in vivo electrophysiological recordings from BLA and PAG neurons and assessed approach-avoidance responses while rats searched for food in the presence of a robotic predator.

The authors observed that rats exhibited a significant increase in the latency to obtain the food pellets and a reduction in the pellet success rate when the predator robot was activated. A subpopulation of PAG neurons showing an increased firing rate in response to the robot activation didn't change their activity in response to food pellet retrieval during the pre- or post-robot sessions. Optogenetic stimulation of PAG neurons increased the latency to procure the food pellet in a frequency- and intensity-dependent manner, similar to what was observed during the robot test. Combining optogenetics with single-unit recordings, the authors demonstrated that photoactivation of PAG neurons increased the firing rate of 10% of BLA cells. A subsequent behavioral test in 3 of these same rats demonstrated that BLA neurons responsive to PAG stimulation displayed higher firing rates to the robot than BLA neurons nonresponsive to PAG stimulation. Next, because the PAG does not project monosynaptically to the BLA, the authors used a combination of retrograde and anterograde neural tracing to identify possible regions that could convey robot-related information from PAG to the BLA. They observed that neurons in specific areas of the paraventricular nucleus of the thalamus (PVT) that are innervated by PAG fibers contained neurons that were retrogradely labeled by the injection of CTB in the BLA. In addition, PVT neurons showed increased expression of the neural activity marker cFos after the robot test, suggesting that PVT may be a mediator of PAG signals to the BLA.

Overall, the idea that the PAG interacts with the BLA via the midline thalamus during a predator vs. foraging test is new and quite interesting. The authors have used appropriate tools to address their questions. However, there are some major concerns regarding the design of the experiments, the rigor of the histological analyses, the presentation of the results, the interpretation of the findings, and the general discussion that largely reduces the relevance of this study.

The authors have fully addressed all my concerns.

---

## [Referee Report · Reviewer #2 (Public Review)]

The authors characterized the activity of the dorsal periaqueductal gray (dPAG) - basolateral amygdala (BLA) circuit. They show that BLA cells that are activated by dPAG stimulation are also more likely to be activated by a robot predator. These same cells are also more likely to display synchronous firing.

The authors also replicate prior results showing that dPAG stimulation evokes fear and the dPAG is activated by a predator.

Lastly, the report performs anatomical tracing to show that the dPAG may act on the BLA via the paraventricular thalamus (PVT). Indeed, the PVT receives dPAG projections and also projects to the BLA. However, the authors do not show if the PVT mediates dPAG to BLA communication with any functional behavioral assay.

The major impact in the field would be to add evidence to their prior work, strengthening the view that the BLA can be downstream of the dPAG.

---

## [Referee Report · Reviewer #3 (Public Review)]

In the present study, the authors examined how dPAG neurons respond to predatory threats and how dPAG and BLA communicate threat signals. The authors employed single-unit recording and optogenetics tools to address these issues in an 'approach food-avoid predator' paradigm. They characterized dPAG and BLA neurons responsive to a looming robot predator and found that dPAG opto-stimulation elicited fleeing and increased BLA activity. Importantly, they found that dPAG stimulation produces activity changes in subpopulations of BLA neurons related to predator detection, thus supporting the idea that dPAG conveys innate fear signals to the amygdala. In addition, injections of anterograde and retrograde tracers into the dPAG and BLA, respectively, along with the examination of c-FOS activity in midline thalamic relay stations, suggest that the paraventricular nucleus of the thalamus (PVT) may serve as a mediator of dPAG to BLA neurotransmission. Of relevance, the study helps to validate an important concept that dPAG mediates primal fear emotion and may engage upstream amygdala targets to evoke defensive responses. The series of experiments provides a compelling case for supporting their conclusions. The study brings important concepts revealing dynamics of fear-related circuits particularly attractive to a broad audience, from basic scientists interested in neural circuits to psychiatrists.

---

## [Author Response]

The following is the authors’ response to the previous reviews.

**Public Reviews**

*Reviewer 1 summarized that: In this revised version of the manuscript, the authors have made important modifications in the text, inserted new data analyses, and incorporated additional references, as recommended by the reviewers. These modifications have significantly improved the quality of the manuscript.*

We are grateful for the reviewer's positive recognition of our revisions.

*Reviewer 2 noted that:*

*(1) The authors do not show if the PVT mediates dPAG to BLA communication with any functional behavioral assay.*

We appreciate the reviewer’s suggestion to include a functional assay to investigate the role of the PVT in mediating communication between the dPAG and BLA. Our primary objective was to confirm the upstream role of the dPAG in processing and relaying naturalistic predatory threat information to the BLA, thereby broadening our current understanding of the dPAG-BLA relationship based on Pavlovian fear conditioning paradigms.

Given previous anatomical findings indicating the absence of direct monosynaptic projections from the dPAG to the BLA (Cameron et al. 1995, McNally, Johansen, and Blair 2011, Vianna and Brandao 2003), we employed both anterograde and retrograde tracers, supplemented by c-Fos expression analysis following predatory threats, to explore possible routes through which threat signals may be conveyed from the dPAG to the BLA. Our findings indicated significant activity within the midline thalamic regions, particularly the PVT as a mediator of dPAG-BLA interactions, corroborating the possibility of dPAGàBLA information flow.

Investigating the PVT's functional role appropriately would require single-unit recordings, correlation analysis of PVT neuronal responses with dPAG and BLA neuronal responses, and pathway-specific causal techniques, involving other midline thalamic regions for controls. This comprehensive study would represent an independent study.

In response to previous feedback, we have carefully revised our manuscript to moderate the emphasis on the PVT's role. Both the Abstract, Results, and Discussion refer more broadly to "midline thalamic regions" and “The midline thalamus” (subheading) rather than specifically to the PVT. In the Introduction, we mention that the PVT "may be part of a network that conveys predatory threat information from the dPAG to the BLA." Our conclusions about the functional interaction between the dPAG and BLA, which broaden the view of Pavlovian fear conditioning, are not contingent on confirming a specific intermediary role for the PVT.

*(2) The author also do not thoroughly characterize the activity of BLA cells during the predatory assay.*

Our previous studies have extensively detailed BLA cell firing characteristics, including their responsiveness to food and/or a robot predator during the predatory assay (Kim et al. 2018, Kong et al. 2021), and compared these findings to other predator studies (Amir et al. 2019, Amir et al. 2015). In the current study, out of 85 BLA cells, 3 were food-specific and 4 responded to both the pellet and the robot, with none of these 7 cells responding to dPAG stimulation.

Given our earlier findings of the immediate responses of BLA neurons to robot activation, we specifically examined whether robot-responsive BLA neurons receive signals from the dPAG. For this analysis, we excluded all food-related cells (pellet cells and BOTH cells) and focused on the time window immediately after robot activation (within 500 ms after robot onset). This approach enabled us to avoid potential confounds from residual effects of robot-induced immediate BLA responses during the animals’ flight and nest entry behaviors.

Furthermore, as previously described, the robot is programmed to move forward a fixed distance and then return, repeatedly triggering foraging behavior. This setup facilitates the analysis of neural changes during food approach and predator avoidance conflicts. However, animals quickly adapt to the robot, reducing freezing and stretch-attend behaviors, making time-stamped analysis of these behaviors unfeasible.

We would like to highlight that the present study explicitly focused on demonstrating whether BLA neurons that responded to intrinsic dPAG optogenetic stimulation also responded to extrinsic predatory robot activation, and compared their firing characteristics to those BLA neurons that did not respond to dPAG stimulation (Figure 3). This targeted analysis provides insights into the responsiveness of BLA neurons to both intrinsic and extrinsic stimuli, furthering our understanding of the dPAG-BLA interaction in the context of predatory threats.

*Reviewer 3 also raised no concerns and stated that: The series of experiments provide a compelling case for supporting their conclusions. The study brings important concepts revealing dynamics of fear-related circuits particularly attractive to a broad audience, from basic scientists interested in neural circuits to psychiatrists.*

We sincerely thank the reviewer for the positive feedback on our revisions.

**Recommendations for the Authors**

*Reviewer 1: There are a few minor concerns that the authors may want to fix:*

*(1) Point (5) The sentence: "The complexity of targeting the dPAG, which includes its dorsomedial, dorsolateral, lateral, and ventrolateral subdivisions" is hard to follow because the ventrolateral subdivision is not part of the dPAG. The authors may want to say specific subregions of the PAG instead. It is also unclear why transgenic animals would be needed for this projection-defined manipulations. The combination of retrograde Cre-recombinase virus with inhibitory opsin or chemogenetic approach may be sufficient.*

We appreciate the reviewer’s insightful feedback regarding our description of the dPAG and the use of transgenic mice in future studies. As suggested, we have corrected the manuscript to exclude the 'ventrolateral' subdivision from the dPAG description, now accurately aligning with pioneering studies (Bandler, Carrive, and Zhang 1991, Bandler and Keay 1996, Carrive 1993) that designated dPAG as including the dorsomedial (dmPAG), dorsolateral (dlPAG) and lateral (lPAG) regions, as cited in our revised manuscript.

We acknowledge the reviewer’s helpful suggestion regarding the use of retrograde Cre-recombinase virus with inhibitory opsins or chemogenetic approaches as viable alternatives. These methods have been incorporated into our discussion (pages 14-15): “While our findings demonstrate that opto-stimulation of the dPAG is sufficient to trigger both fleeing behavior and increased BLA activity, we have not established that the dPAG-PVT circuit is necessary for the BLA’s response to predatory threats. To establish causality and interregional relationships, future studies should employ methods such as pathway-specific optogenetic inhibition (using retrograde Cre-recombinase virus with inhibitory opsins; Lavoie and Liu 2020, Li et al. 2016, Senn et al. 2014) or chemogenetics (Boender et al. 2014, Roth 2016) in conjunction with single unit recordings to fully characterize the dPAG-PVT-BLA circuitry’s (as opposed to other midline thalamic regions for controls) role in processing predatory threat-induced escape behavior. If inactivating the dPAG-PVT circuits reduces the BLA's response to threats, this would highlight the central role of the dPAG-PVT pathway in this defense mechanism. Conversely, if the BLA's response remains unchanged despite dPAG-PVT inactivation, it could suggest the existence of multiple pathways for antipredatory defenses.”

This revision addresses the critique by clarifying the anatomical description of the dPAG and emphasizing the feasibility of using targeted viral approaches without the necessity for transgenic animals.

*(2) Point (6e) The authors mentioned that "pellet retrieval" was indicated by the animal entering a designated zone 19 cm from the pellet, driven by hunger. Entering the area 19cm of distance should be labeled as food approaching rather then food retrieval because in many occasions the animals may be some seconds away of grabbing the pellet.*

We agree and incorporate the change (pg. 22).

*(3) Point (11) We would strongly recommend the authors to replace the terminology "looming" by "approaching" to avoid confusion with several previous studies looking at defensive behaviors in responses to looming induced by the shadow of an object moving closer to the eyes.*

Done.

*(4) Point (17) The authors mentioned that "A total of three rats were utilized for the robot testing experiments depicted in Fig. 2 G-J." However, the figure indicates a total of 9 ChR2 and 4 controls.*

We apologize for the confusion in our previous author responses. To examine the optical stimulation effects on behavior in Fig. 2G-J, we used a total of 9 ChR2 and 4 EYFP rats. The experimental sequence is detailed in the previously revised manuscript (pg. 20): “For optical stimulation and behavioral experiments, the procedure included 3 baseline trials with the pellet placed 75 cm away, followed by 3 dPAG stimulation trials with the pellet locations sequentially set at 75 cm, 50 cm, and 25 cm. During each approach to the pellet, rats received 473-nm light stimulation (1-2 s, 20-Hz, 10-ms width, 1-3 mW) through a laser (Opto Engine LLC) and a pulse generator (Master-8; A.M.P.I.). Additional testing to examine the functional response curves was conducted over multiple days, with incremental adjustments to the stimulation parameters (intensity, frequency, duration) after confirming that normal baseline foraging behavior was maintained. For these tests, one parameter was adjusted incrementally while the others were held constant (intensity curve at 20 Hz, 2 s; frequency curve at 3 mW, 2 s; duration curve at 20 Hz, 3 mW). If the rat failed to procure the pellet within 3 min, the gate was closed, and the trial was concluded.”

This clarification ensures that the actual number of animals used is accurately reflected and aligns with the figure data, addressing the reviewer's concern.

*Reviewer 2: The authors made important changes in the text to address study limitations, including citations requested by the Reviewers and additional discussions about how this work fits into the existing literature. These changes have strengthened the manuscript.*

*(1) However, the authors did not perform new experiments to address any of the issues raised in the previous round of reviews. For example, they did not make optogenetic manipulations of the pathway including the PVT, and did not add any loss of function experiments. The justification that these experiments are better suited for future reports using mice is not convincing, because hundreds of papers performing these types of circuit dissection assays have been performed in rats.*

We appreciate the reviewer's comments regarding the experimental scope of our study. Our study’s primary objective was to explore the dPAG’s upstream functional role in processing and conveying naturalistic predatory threat information to the BLA, extending our current understanding of the dPAG-BLA relationship based on Pavlovian fear conditioning paradigms. We believe that our findings effectively address this goal.

Our use of anterograde and retrograde tracers, supplemented by c-Fos expression analysis in response to predatory threats, was primarily conducted to verify the possibility of the dPAGàBLA information flow during predator encounters. This involved exploring potential routes through which threat signals might be conveyed from the dPAG to the BLA, given the lack of direct monosynaptic projections from the dPAG to BLA neurons (Cameron et al. 1995, McNally, Johansen, and Blair 2011, Vianna and Brandao 2003). This methodology helped us identify a potential structure, PVT, for more in-depth future studies. A thorough examination of the PVT's role would require single-unit recordings and causal techniques, incorporating other midline thalamic regions as controls, representing a significant and separate study on its own.

In response to prior feedback, we have carefully revised our manuscript to generally address the role of "midline thalamic regions" rather than focusing specifically on the PVT. We wish to emphasize that our findings, which illustrate unique functional interactions between the dPAG and BLA in response to a predatory imminence, remain compelling and informative even without definitive evidence of the PVT’s involvement.

*Reviewer 3: In the revised version of the manuscript, the authors addressed adequately all the concerns raised by the reviewers.*

We thank the reviewer for the thoughtful feedback on the earlier version of our manuscript and for reexamining the revisions we have made.

References

Amir, A., P. Kyriazi, S. C. Lee, D. B. Headley, and D. Pare. 2019. "Basolateral amygdala neurons are activated during threat expectation." *J Neurophysiol* 121 (5):1761-1777.

Amir, A., S. C. Lee, D. B. Headley, M. M. Herzallah, and D. Pare. 2015. "Amygdala Signaling during Foraging in a Hazardous Environment." *J Neurosci* 35 (38):12994-3005.

Bandler, R., P. Carrive, and S. P. Zhang. 1991. "Integration of somatic and autonomic reactions within the midbrain periaqueductal grey: viscerotopic, somatotopic and functional organization." *Prog Brain Res* 87:269-305.

Bandler, R., and K. A. Keay. 1996. "Columnar organization in the midbrain periaqueductal gray and the integration of emotional expression." *Prog Brain Res* 107:285-300.

Boender, A. J., J. W. de Jong, L. Boekhoudt, M. C. Luijendijk, G. van der Plasse, and R. A. Adan. 2014. "Combined use of the canine adenovirus-2 and DREADD-technology to activate specific neural pathways in vivo." *PLoS One* 9 (4):e95392.

Cameron, A. A., I. A. Khan, K. N. Westlund, and W. D. Willis. 1995. "The efferent projections of the periaqueductal gray in the rat: a Phaseolus vulgaris-leucoagglutinin study. II. Descending projections." *J Comp Neurol* 351 (4):585-601.

Carrive, P. 1993. "The periaqueductal gray and defensive behavior: functional representation and neuronal organization." *Behav Brain Res* 58 (1-2):27-47.

Kim, E. J., M. S. Kong, S. G. Park, S. J. Y. Mizumori, J. Cho, and J. J. Kim. 2018. "Dynamic coding of predatory information between the prelimbic cortex and lateral amygdala in foraging rats." *Sci Adv* 4 (4):eaar7328.

Kong, M. S., E. J. Kim, S. Park, L. S. Zweifel, Y. Huh, J. Cho, and J. J. Kim. 2021. "'Fearful-place' coding in the amygdala-hippocampal network." *Elife* 10.

Lavoie, A., and B. H. Liu. 2020. "Canine Adenovirus 2: A Natural Choice for Brain Circuit Dissection." *Front Mol Neurosci* 13:9.

Li, Y., L. Hickey, R. Perrins, E. Werlen, A. A. Patel, S. Hirschberg, M. W. Jones, S. Salinas, E. J. Kremer, and A. E. Pickering. 2016. "Retrograde optogenetic characterization of the pontospinal module of the locus coeruleus with a canine adenoviral vector." *Brain Res* 1641 (Pt B):274-90.

McNally, G. P., J. P. Johansen, and H. T. Blair. 2011. "Placing prediction into the fear circuit." *Trends Neurosci* 34 (6):283-92.

Roth, B. L. 2016. "DREADDs for Neuroscientists." *Neuron* 89 (4):683-94.

Senn, V., S. B. Wolff, C. Herry, F. Grenier, I. Ehrlich, J. Grundemann, J. P. Fadok, C. Muller, J. J. Letzkus, and A. Luthi. 2014. "Long-range connectivity defines behavioral specificity of amygdala neurons." *Neuron* 81 (2):428-37.

Vianna, D. M., and M. L. Brandao. 2003. "Anatomical connections of the periaqueductal gray: specific neural substrates for different kinds of fear." *Braz J Med Biol Res* 36 (5):557-66.